First report of leopard fossils from a limestone cave in Kenting area, southern Taiwan

Chi Tzu-Chin 1 2
Gan Yi 3
Yang Tzu-Ruei tzurueiyang@nmns.edu.tw 1 3
Chang Chun-Hsiang cch@nmns.edu.tw 1
1 Department of Geology, National Museum of Natural Science , Taichung , Taiwan
2 Department of Earth and Environmental Sciences, National Chung Cheng University , Chiayi , Taiwan
3 Department of Earth Sciences, National Cheng Kung University , Tainan , Taiwan
Nganvongpanit Korakot
Electronic publication date: 2021 Aug 23
Publication date: 2021
Volume: 9
Electronic Location ID: e12020
Received 2020 Aug 18; Accepted 2021 Jul 29
Copyright: ©2021 Chi et al.
Copyright year: 2021
Copyright holder: Chi et al.
License: This is an open access article distributed under the terms of the Creative Commons Attribution License, which permits unrestricted use, distribution, reproduction and adaptation in any medium and for any purpose provided that it is properly attributed. For attribution, the original author(s), title, publication source (PeerJ) and either DOI or URL of the article must be cited.
License URL: https://creativecommons.org/licenses/by/4.0/

Keywords: Felidae, Morphology, Panthera pardus, Late Pleistocene, Geometric morphometric analysis

Funding: Ministry of Science and Technology (MOST), Taiwan 104-2116-M-178-005 106-2116-M-178-001 108-2116-M-178-003-MY2 1110-2116-M-178-002 This study was supported by the grants of the Ministry of Science and Technology (MOST), Taiwan (104-2116-M-178-005 and 106-2116-M-178-001 to Chun-Hsiang Chang and 108-2116-M-178-003-MY2 and 110-2116-M-178-002 to Tzu-Ruei Yang). The funders had no role in study design, data collection and analysis, decision to publish, or preparation of the manuscript.

==============================
Longshia-dong Cave, a limestone cave located in the Kenting area within the Kenting National Park of southern Taiwan, yields numerous terrestrial mammalian fossils. Many of them were not reported in historical literature and are neither present in Taiwan. For instance, no historical literature mentioned leopards inhabited in Taiwan, and thus their existence remained unknown. This study describes three fossil leopard (Panthera pardus) teeth uncovered from Longshia-dong Cave. Two isolated lower premolars and one lower molar, respectively p3, p4 and m1, were discovered in a very small area (11 × 6 cm) and show a series of progressive increase in size. Thus, the three teeth should have been belonging to the same individual from the subfamily of Pantherinae. Traditional linear measurements and two-dimensional geometric morphometric analysis for the occlusal surface outlines were conducted on the fossil teeth and extant pantherines inhabited in Asia such as clouded leopards (Neofelis nebulosa), leopards (Panthera pardus), and tigers (Panthera tigiris). Results show that the fossil teeth are similar both in size and morphology to the teeth of extant leopards, suggesting the assignment of the fossil teeth to leopards. This study, for the first time, reported the presence of leopards in the Late Pleistocene of Taiwan. In addition, the smaller size of the fossil teeth in comparison with Chinese fossil leopards is putatively attributed to insular dwarfism or individual size variability, yet more studies are required.

Introduction

Longshia-dong Cave (= Lobster Cave) is located in the Kenting Forest Recreation Area (KFRA) of the Kenting National Park, southernmost Taiwan (Fig. 1). The KFRA is covered with thick limestone (Hengchun Limestone) that deposited during the Middle Pleistocene (Gong & Yui, 1998). A number of caves and fissures formed in the Hengchun Limestone, some of which, such as Longshia-dong Cave, accumulated numerous terrestrial mammal fossils. This cave opens at ca. 240 m above the present sea level and is a small tunnel inclined gently toward its inner part (Kawamura, Chang & Kawamura, 2016).

Figure 1 Location and geological map of the discovery site of the leopard fossils in Taiwan.

A map of Taiwan shows the location of the Kenting National Park (A), and a geological map of southernmost Taiwan is presented in (B). The location of the discovery site, Longshia-dong Cave, is indicated by a red star in (B).

The investigation of Longshia-dong Cave was initiated by Prof. Yoshinari Kawamura from the Aichi University of Education, Japan and Dr. Chun-Hsiang Chang from the National Museum of Nature Science, Taiwan (Kawamura, Chang & Kawamura, 2016). To date, fossils of Cervidae, Rodentia (e.g., Microtus and Hystrix), Carnivora (three teeth; this study), Macaca sp., and Rhinolophus sp. uncovered from this cave are identified (Kawamura, Chang & Kawamura, 2016). However, the Microtus (Rodentia) is now restricted to high mountains in Taiwan and the Hystrix sp. (Rodentia) is no longer present in Taiwan. Besides, leopard teeth were not reported in Taiwan previously. These lines of evidence indicate a very different faunal setting in comparison with the present one.

In previous excavations scientists have discovered putative Middle-Late Pleistocene big cat fossils, such as Panthera cf. tigris and Panthera sp. from the Chochen area in southern Taiwan (Fig. 1; Otsuka, 1984; Chen, 2000a; Chen, 2000b; Wei, 2007). Besides, fossil remains of Panthera tigris have also been collected from the Penghu Channel, a N-S striking submarine valley off the west coast of Taiwan (Fig. 1) (Ho, Qi & Chang, 1997; Asahara et al., 2015). While both of these studies indicate a rich fossil record of felids in the Middle-Late Pleistocene of Taiwan, the modern Felidae in Taiwan are only featured by two species, including clouded leopard (Neofelis nebulosa) and leopard cat (Prionailurus bengalensis). Even more recently, Chiang et al. (2015) presumed that the former is extinct in Taiwan, thus leaving only the latter present to our knowledge.

Previous studies indicated the similarities of Taiwan’s fauna to the Early Pleistocene fauna of southern China (e.g., Otsuka & Lin, 1984; Lai, 1989; Qi, Ho & Chang, 1999; Chen, 2000a; Fooden & Wu, 2001). The fauna of southern China probably entered Taiwan in the Late Pliocene to Early Pleistocene, when Taiwan was connected with China (Lai, 1989; Chen, 2000a; Fooden & Wu, 2001); however, only clouded leopard was reported in Taiwan’s historical record. While Swinhoe (1862) had never seen a living individual, he described it as a small, short-tailed, small-footed animal based on the fur specimen and named it as Leopardus brachyurus. Later, Swinhoe (1870) replaced L. brachyurus with Felis macrocelis, but L. brachyurus is still the most commonly used name.

In the excavation to Longshia-dong Cave in 2014, three teeth were collected from the same horizon. A preliminary study has indicated their affinity to feline remains, though further investigations are needed (Gan, 2016). This study thus aims to reveal their taxonomic affinity based on traditional linear and geometric measurements. Besides, the comparison between the studied material and leopard fossils from various sites of the Pleistocene in China, including the Zhoukoudian site (Teilhard de Chardin & Pei, 1941) and Mentougou Bull Eye Cave (Deng, Huang & Wang, 1999) of Beijing, Lantian (Gongwangling) of Shanxi (Hu & Qi, 1978), Anyang (Yinxu) of Henan Province (Teilhard de Chardin & Young, 1936), Liucheng Cave of Guangxi Province (Pei, 1987), also allows us to reveal more details.

Geological setting

The Kenting National Park is located in the Hengchun Peninsula, southernmost Taiwan (Fig. 1). The Hengchun Peninsula represents the earliest stage of the Taiwanese orogeny (Huang, Cheng & Jeh, 1985), and therefore many incipient thrust faults are observed. A major boundary fault, the N-S striking Hengchun Fault (Fig. 1; Chen et al., 2005), divides the Hengchun Peninsula into two terranes, including the Central Range in the East and the Western Foothills (Hengchun Valley and West Hengchun Hill) in the West (Yen & Wu, 1986). Since the Late Pleistocene, the Hengchun Peninsula was uplifted at a rate of 2–6 mm/yr (Chen et al., 2005) and thus gave rise to the development of coral reef and limestone (Hengchun Limestone). Many karst caves were afterward formed and harbor various fossils. A thin layer of reddish sand and gravels (Hsu, 1989; Hseu et al., 2004) overlying the Hengchun Limestone was named as Eluanpi Bed in the Southeast, or Taiping Formation in the West, to Longshia-dong Cave (Fig. 1).

Our studied area, Longshia-dong Cave, is located in the southeastern part of the Kenting National Park, a national park that is featured by the karst landscape mostly contributed by the Hengchun Limestone (Fig. 1). The cave opens to the east, measuring 30–40 m long, 8 m wide, and about 5 m deep, and a puddle was found in the end of the cave (Kawamura, Chang & Kawamura, 2016). Most of the limestone in the cave is covered by a layer of reddish sediment composed of limestone pebbles and fossils, though the boundary between the reddish sediment and Hengchun Limestone is unclear. A flow stone made of carbonate calcite was found 7 m away from the cave entry, and the fossil-bearing sediments are found behind it. The fossil-bearing sediments are characterized by a mixture of reddish sand and mud, as well as limestone pebbles, which shows a great similarity to the Eluanbi Bed. Besides, a layer of blackish mud and light yellowish-red silt (possibly loess) overlying the fossil-bearing sediments, based on a previous study using 14C dating, is composed of recent alluvial deposits since the Holocene (Wang, 2015).

Since an estimation of the age for the fossil-bearing sediments was inaccessible, previous studies have put emphasis on the age of formation of the Hengchun Limestone, which gave a maximum estimation for the leopard fossils uncovered from the cave. The Hengchun Limestone was either considered two-stage (Gong, 1982) or three-stage (Shih et al., 1989) reef formation, but both studies have concluded a dating result of 500 ka, which is similar to the result arisen from nannofossils (NN19, around 500 ka; Chi, 1982). Besides, another geochemical chronological study has suggested a date of 325–125 ka for the age of formation of the Hengchun Limstone (Gong, 1994).

Afterwards, Kawamura, Chang & Kawamura (2016) offered the first estimation of the age for the fossil-bearing sediments. Based on stratigraphic relationships, Kawamura, Chang & Kawamura (2016) suggested an age of the Late Pleistocene (126 to 12 ka) for the fossil-bearing sediments, which is much older than the 7 ka revealed by 14C dating of the bone collagen extracted from some ruminant antlers and lower jaws uncovered from the same locality. Because of the rarity and poor-preservation of the leopard remains in this study, a rigorous age of the leopard remains is inaccessible. While the age of the fossils requires further investigations, this study will discuss further details of the leopard remains based on the temporal range from 500 to 12 ka.

Materials & Methods

Three well-preserved whitish fossil teeth were collected from Longshia-dong cave in the 2014 excursion led by one of the authors (C.-H. Chang). All of them (F056584, F056585, and F056586) are housed at the National Museum of Natural Science, Taichung, Taiwan (NMNS) under the accession numbers provided. Before excavation, we used strings to divide the cave ground into a grid contributed by mostly 50 × 50 cm areas. The three teeth were discovered in a small portion (11 × 6 cm) out of one of the areas and present a series of progressive increase in size; thus, we believed they should have been belonging to a felid individual (Fig. 2). The felid fossil teeth, based on their shape (Hillson, 2005), are assignable to p3 (F056584), p4 (F056585) and m1 (F056586) from the right mandible (Fig. 2). In addition to the three felid teeth, the skulls and mandibles of 17 extant specimens, including seven clouded leopards (Neofelis nebulosa, but one of them might be Sunda clouded leopard (Neofelis diardi)), five leopards (Panthera pardus), and five tigers (Panthera tigris), which are housed at the NMNS, Endemic Species Research Institute, and Taipei Zoo, respectively, were also included in this study (see the details in the Table S1). Prior to our qualitative studies of the fossil and extant felid teeth (Kawamura, 1992; Fukawa, 2000), we compare their morphological features to determine the assignment of the fossil felid teeth (Fig. 3). Photos of all specimens were taken with a Panasonic Lumix DMC-GF1 camera and a Panasonic Lumix GF1 14–45 mm/F3.5−5.6 lens. For the traditional linear measurements and geometric morphometric studies, these photos were afterwards imported into the tpsDig 2.05 (Rohlf, 2005).

Figure 2 The three felid lower cheek teeth from Longshia-dong Cave.

The three felid lower cheek teeth from Longshia-dong Cave, including p3 (a, F056584), p4 (b, F056585), and m1 (c, F056585), and their buccal views (1), lingual views (2), and occlusal views (3).

Figure 3 A comparison of the teeth (p3, p4, and m1) of three pantherines.

A comparison of the teeth (p3, p4, and m1) of three pantherines including (A) clouded leopard, (B) leopard, and (C) tiger. Three significant characters are indicated by (1) the developmental level of p3 paraconid, (2) the shape of p4 occlusal surface (the difference between paraconid and protoconid widths), and (3) the size difference between m1 paraconid and m1 protoconid.

Traditional linear measurements were taken point-to-point; a total of 22 dental dimensions including antero-posterior crown length (1, 5, 15 in Fig. 4), dorsoventral crown height (3, 7, 10, 13, 17, 20 in Fig. 4), width of each cusp (4, 8, 11, 14, 18, 21 in Fig. 4), anteroposterior length of cusps (2, 6, 9, 12, 16, 19 in Fig. 4), and crown height at the place of carnassial notch (22 in Fig. 4) (Christiansen, 2008), were obtained from all specimens if accessible (a fossil individual and 17 extant specimens, see the details in the Table S1). The three fossil felid teeth belonged to the right mandible and therefore 22 dental dimensions were obtained. The 17 extant specimens, on the other hand, permit the measurement of both lower right and left jaws, thus contributing to a 17 × 22 × 2 data matrix (note that some dental dimensions are not available due to poor preservation, see the details in the Table S1).

Figure 4 Measurements of p3, p4, and m1.

Measurements of p3, p4, and m1 from the buccal side (A, from left to right) and from the occlusal side (B, from left to right). For the numbered dimensions, see Table 1.

The data from the aforementioned traditional linear measurements were introduced into two rounds of principal component analysis (PCA) (Morrison, 1976; Dunteman, 1994), which plots the data to a new coordinate system. The new coordinate system is contributed by N-1 principal components, which are orthogonal to each other (Jolliffe, 2002; Hsu, 2003). All PCA in this study were performed with the R package “stats” (R Core Team, 2013). In the first round of PCA, 22 dental dimensions from the three teeth (p3, p4, and m1) of a mandible (either right or left) are seen as a dataset. Any missing of the 22 dental dimensions from p3, p4, or m1 leads to the removal of the whole dataset from the first round of PCA. Ultimately, one dataset from the felid fossil, ten datasets from the seven clouded leopards, five datasets from the five leopards, and nine datasets from the five tigers, were used in the first round of PCA (Fig. 5A).

Figure 5 Two rounds of PCA analysis of the traditional linear measurements.

The first round of PCA analysis based on all datasets is presented in (A). The second round of PCA analysis was performed for p3 (B), p4 (C), and m1 (D), separately. The result from the three ratios converted from four p3 numbered dimensions (A: 2/1, B: 3/1, C: 2/3) is shown in (B). The result from the six ratios converted from seven p4 numbered dimensions (A: 6/ 5, B: 7/5, C: 9/5, D: 10/5, E: 12/5, F: 13/5 is presented in (C). The result from the seven ratios converted from eight m1 numbered dimensions (A: 16/15, B: 17/15, C: 19/15, D: 20/15, E: 16/19, F: 17/20, G: 22/15 is shown in (D). See the details of the numbered dimensions in Fig. 3 and Table 1. Red triangles, Panthera tigris; yellow circles, Panthera pardus; cyan square, Neofelis nebulosa and Neofelis diardi; black cross, the studied fossil felid teeth. Red arrow represents a simplified ratio that shows the trend relating to the two principal components. Factor loadings of each principal component is indicated by the arrow length.

To avoid the effect of the interspecific difference in tooth size on the first round of PCA, we performed the second round of PCA with the R package “stats” for p3, p4, and m1, separately. For instance, we obtained three ratios, including of protoconid length (2 in Fig. 4 and Table 1) to crown length (1 in Fig. 4 and Table 1), of protoconid height (3 in Fig. 4 and Table 1) to crown length (1 in Fig. 4 and Table 1), and protoconid length (2 in Fig. 4 and Table 1) to protoconid height (3 in Fig. 4 and Table 1) based on the data from all available p3 (a fossil felid, 13 clouded leopards, five leopards, and nine tigers). The three ratios were used for PCA analysis with the package “stats.” For PCA analysis (Fig. 5B), on the other hand, six ratios and seven ratios were obtained based on the data from all available p4 and m1, respectively (see the details in Fig. 5) and were analyzed with the package “stats” (Figs. 5C and 5D). Besides, we construct a bivariate plot of the selected numbered dimensions of p3, p4, m1 (1, 4, 5, 11, 15, 21 from Table 1) of the fossil and extant leopard specimens with the regression lines for each selected number dimensions (Fig. 6).

Table 1 Selected dimensions of teeth.

dimension		
1	p3 crown length	
2	p3 protoconid length	
3	p3 protoconid height	
4	p3 protoconid crown width	
5	p4 crown length	
6	p4 paraconid length	
7	p4 paraconid crown height	
8	p4 paraconid crown width	
9	p4 potoconid length	
10	p4 protoconid crown height	
11	p4 protoconid width	
12	p4 hypoconid length	
13	p4 hypoconid height	
14	p4 hypoconid crown width	
15	m1 crown length	
16	m1 paraconid length	
17	m1 paraconid height	
18	m1 paraconid width	
19	m1 protoconid length	
20	m1 protoconid height	
21	m1 protoconid width	
22	m1 carnassial notch height	

Figure 6 A bivariate plot of selected numbered dimensions of p3, p4, m1 of the fossil and extant leopard specimens.

A bivariate plot of selected numbered dimensions (1, 4, 5, 11, 15, 21 from Table 1. of p3, p4, m1 of the fossil and extant leopard specimens (in mm). Dotted light blue, black, dark blue, green, and purple lines represent the regression lines for the dimensions of the five extant specimens, including the right and left sides of the specimen no. 1348-1, the right and left sides of the specimen no. 1431-1, and the left side of the specimen no. 549-1 (see Table S1). The regression line for the dimensions of the fossil specimen is marked in red. Triangles, p3; squares, p4; circles, m1.

In addition to the PCA analyses based on traditional linear measurements, we also performed geometric morphometric analysis (Slice, 1996) because of its utility of revealing the morphological differences between different groups (Zelditch, Fink & Swiderski, 1995) and the ability to exclude the factor of size. All photographs were input into the program tpsUtil for building up a tps file. To access the morphology of the occlusal surface of each tooth in the absence of apparent landmarks, we used the “curve mode” in the program tpsDig 2.05 (accessed on Dec 1, 2014 from http://life.bio.sunysb.edu/morph/; (Rohlf, 2005) to place evenly distributed 150 semi-landmarks around the occlusal surface on each photo (Gunz & Mitteroecker, 2013). The 150 semi-landmarks were then digitized from photographs using tpsDig 2.05, which converted the points marked on the photographs into Cartesian x, y coordinates. After scaling and alignment of the digitized semi-landmarks using generalized Procrustes analysis, a relative warp analysis (RWA) was then performed on the set of specimen semi-landmarks in tpsRelw (Rohlf, 2007) to unravel the morphological variation between the fossil and extant teeth. We then visualize the morphological variation from RWA by plotting the relative warp axes as a PCA. Relative differences are presented in the form of thin plate spline deformation grids (Figs. 7–9; Zelditch, Lundrigan & Garl Jr, 2004; Tseng, Wen & Chen, 2010).

Figure 7 The shape variation of the occlusal surface of p3 in three pantherines.

The shape variation of the occlusal surface of p3 in three pantherines, as revealed by a principal components analysis of three warp scores. (A) RW1 versus RW2; (B) RW1 versus RW3; (C) RW2 versus RW3. RW1 in a positive direction explains 50.54% variance, RW2 in a positive direction explains 15.02% variance, and RW3 in a positive direction explains 9.25% variance. Red triangles, Panthera tigris; yellow circles, Panthera pardus; cyan square, Neofelis nebulosa and Neofelis diardi ; black cross, the studied fossil felid teeth.

Figure 8 The shape variation of occlusal surface of p4 in three pantherines.

The shape variation of occlusal surface of p4 in three pantherines, as revealed by a principal components analysis of warp scores. Plot of (A) RW1 versus RW2; (B) RW1 versus RW3; (C) RW2 versus RW3. Red triangles, Panthera tigris; yellow circles, Panthera pardus; cyan square, Neofelis nebulosa and Neofelis diardi; black cross, the studied fossil felid teeth. (D) the shape of fossil p4, (E) thin-plate spline deformation grid depicting shape variation along RW1 in a positive direction explains 46.01% variance, (F) thin-plate spline deformation grid depicting shape variation along RW2 in a positive direction explains 24.52% variance, (G) thin-plate spline deformation grid depicting shape variation along RW3 in a positive direction explains 9.05% variance.

Results

Description and morphological comparison

Family Felidae Fischer von Waldheim (1817)	
Subfamily Pantherinae Pocock, 1917	
Genus PantheraOken, 1816	
Panthera pardus (Linnaeus, 1758) (Fig. 2)	

Based on our observation of all extant felid specimens and previous studies (Gray, 1867; Christiansen & Kitchener, 2011; King, 2012), we concluded the following common dental characters (Fig. 3): (1) two-rooted p3, p4 and m1; (2) p3 is smaller than p4; (3) p4 is in a similar size to m1; (4) p3, p4 has three cusps (paraconid, protoconid and hypoconid) and its paraconid and hypoconid are well-developed (5) m1 has two well-developed cusps (paraconid and protoconid) and undeveloped talonid. All of the aforementioned features are present in the fossil teeth and thus indicate the assignment of the fossil teeth to Felidae. Moreover, the fossil teeth can be further assigned to Pantherinae based on their similar size to extant pantherines. Pumas (Puma concolor), which belong to Felinae, a sister group to Pantherinae, are the only group of felines that present similar dental size; however, their current distribution (only in Americas) makes the attribution of the fossil teeth to puma unlikely.

In previous studies (Meachen-Samuels & Van Valkenburgh, 2009), dental size is the only character that was used for distinguishing different pantherines, such as lions (Panthera leo), snow leopards (Panthera uncia), tigers, clouded leopards, and leopards. For instance, the dental size of lions is significantly larger than the one of the fossil teeth. Moreover, the attribution of the fossil teeth to snow leopards is unlikely since snow leopards are only present in high mountains. However, dental size is not an indicator for the distinguishment between tigers, clouded leopards, and leopards. This study, however, shows that several pantherines, including clouded leopards, leopards, and tigers, can be distinguished based on three dental characters, including (1) the level of p3 paraconid development, (2) the shape of p4 occlusal surface, (3) the size difference between m1 paraconid and m1 protoconid (Fig. 3; Table 2).

Table 2 Morphological comparisons of three extant species and the fossil. ○, present; –, absent; △, uncertain.

	p3 paraconid	m1 metaconid	size comparison between the paraconid, and the protoconid, of m1	chubby talonid on p3 and p4	occlusal surface	
Longshia-dong Cave fossils	○	–	similar	○	intermediate	
Modern clouded leopards (Neofelis sp.)	–	△	bigger protoconid	–	narrow	
Modern tigers (Panthera tigris)	○	–	similar	○	intermediate	
Modern leopards (Panthera pardus)	△	–	similar	○	chubby	

Clouded leopards have two distinct features, including an absent paraconid of p3 (as seen in all examined clouded leopards) and a well-developed protoconid of m1, while both of which are absent in the fossil teeth from Longshia-dong Cave (Fig. 2). Thus, the assignment of the fossil teeth from Longshia-dong Cave to clouded leopards is here excluded. On the other hand, tiger teeth are characterized by a highly developed p4 paraconid, which is not seen in the fossil teeth (Figs. 2 and 7). Moreover, the p3 of tigers has a lower paraconid than the one of the fossil (Figs. 2 and 7). Tigers also present a wider crown in all teeth, especially in m1, than the ones of the fossil (Table 2).

The fossil pantherine teeth, in addition to their size, show many distinct features that are similar to those of the extant leopards, such as the presence of p3 paraconid and the shape of occlusal surface. The presence of m1 and slightly worn enamel further indicate that the fossil teeth might belong to a very young adult (Stander, 1997). The presence of hollow, not fully developed roots in the fossil teeth further suggests a juvenile origin (about 1–2 years old). However, the gender of the fossil pantherine is uncertain because of the absence of canine teeth (whose size is a key difference between male and female carnivores), the lacking of morphological differences between male and female leopard teeth (Pocock, 1930), and the extremely poor sample size MNI = 1 (minimum number of individuals).

Traditional linear measurement

While our aforementioned morphological comparison indicates the affinity of the fossil teeth to leopards (Panthera pardus), traditional linear measurement was performed for further lines of evidence. The result of the first round of PCA based on the 22 dental dimensions shows a significant disparity between tigers and the others (clouded leopards and leopards) (Fig. 5A) (factor loading of each component shows on Table S3), although the disparity between clouded leopards and leopards is inapparent. In the second round of PCA based on various ratios from the dimensions of p3, p4, and m1, no pattern is indicated by the result of the PCA on p3 (Fig. 5B) and the fossil teeth is not clustered by any group. However, based on the result of PCA on p4 and m1 (Figs. 5C and 5D), the assignment of the fossil teeth to clouded leopards is excluded.

While the above PCAs have potentially rules out the affinity of the fossil teeth to leopards, we further conducted a comparison of the selected dental dimensions including 1, 4, 5, 11, 15, 21 (Table 1) between the fossil and five extant leopard specimens (Fig. 6). The result shows a similar covariation of the p3, p4, and m1 between the fossil and five extant leopard specimens, further supporting our previous inference that the three fossil teeth belonged to the same individual.

Geometric morphometric analysis

In addition to the traditional linear measurement (Fig. 5), geometric morphometric analysis was also performed for further lines of evidence. In the RWA based on all third premolars (p3), the first three relative warp axes accounted for 74.81% of the total variation, though no morphological disparities were revealed (Fig. 7). The RWA based on all fourth premolars (p4), however, shows the disparity between clouded leopards and the others in the plots of RW1 to RW2 (Fig. 8A) and RW2 to RW3 (Fig. 8C). The RW2 explains 24.52% of the total shape variance and relates primarily to the prominence of the mesial side toward buccal or lingual side (Fig. 8F). On the other hand, the RWA based on all first molars indicates the isolation of tigers in the plots of RW1 to RW2 (Fig. 9A) and RW2 to RW3 (Fig. 9C). Striking samely, the RW2 in the RWA based on all first molars, which accounts for 22.67% of the total variation, is also the best indicator that excludes tigers.In summary, the fossil pantherine teeth uncovered from Longshia-dong Cave, based on various lines of evidence, are assignable to leopards. While the plot of p3, p4, and m1 sizes of the fossil and extant leopards indicates that only the fossil m1 is encompassed within the variations of the extant leopard teeth, the slope of the regression line based on the three fossil teeth is similar to the ones of those based on the extant specimens. Thus, the three fossil teeth were very likely from a common origin.

Figure 9 The shape variation of occlusal surface of m1 in three pantherines.

The shape variation of occlusal surface of m1 in three pantherines, as revealed by a principal components analysis of warp scores. (A) RW1 versus RW2; (B) RW1 versus RW3; (C) RW2 versus RW3.Red triangles, Panthera tigris; yellow circles, Panthera pardus; cyan square, Neofelis nebulosa and Neofelis diardi; black cross, the studied fossil felid teeth. (D) the shape of fossil p4; (E) thin-plate spline deformation grid depicting shape variation along RW1 in a positive direction which explains 59.58% variance, (F) thin-plate spline deformation grid depicting shape variation along RW2 in a positive direction explaines 22.6% variance; (G) thin-plate spline deformation grid depicting shape variation along RW3 in a positive direction explains 4.77% variance.

Discussion

An integrative, qualitative method for the identification of leopard teeth

Morphological comparisons have been utilized in many studies for the species identification of various mammalian teeth, such as elephants (Todd, 2010), hedgehogs (Gould, 2001) and moles (Van Cleef-Roders & Van Den Hoek Ostende, 2001). Despite the similarity both in size and morphology of pantherine teeth (lions excluded as their teeth are apparently larger than leopard teeth), several studies offered qualitative methods on the basis of the size and morphology for the identification of the subfamily of Pantherinae (e.g., Hemmer, 1966). On the other hand, canid teeth are much more complex than felid teeth and thus have more homologous features for landmarks. Therefore, Asahara et al. (2015) have applied morphological comparisons in combination with geometric morphometric analyses to canine teeth, which is an integrative method that has never been performed on felid teeth. This study, for the first time, demonstrates that an integration of morphological comparisons, linear measurements, and geometric morphometric studies allows species identification of the subfamily of Pantherinae based on teeth. Based on the integrated approach, we thus assign the studied fossil teeth (F056584, F056585, and F056586) to Panthera pardus.

Taphonomic implications for an adult origin of the fossil teeth

A previous study by Stander (1997) suggested that tooth wear in leopards is a reliable indicator for their ages. The eruption of permanent teeth in one to two years old leopards is completed and thus are whitish and wearless. The crown tips are worn in individuals older than three years old, and the tooth wear appears first on incisors, then on canines, on premolars, and finally on molars. The wear is apparent on the teeth of the individuals of five or six years old. Although the fossil teeth in this study only preserve p3, p4, and m1, their complete eruption indicates an age older than one year. Moreover, the slight wear on their crown tips offers further information that the fossil leopard should have been younger than five years old at death.

A behavioral or sedimentary origin?

Extant leopards tend to carry their prey to a safe, isolated location for storage (De Ruiter & Berger, 2000). Many studies, based on field observation, indicated that the leopards in South Africa prefer carrying their prey into their caves, thus contributing to the large number of skeletal remains in many caves in South Africa (Le Roux & Skinner, 1989). In addition to the fossil leopard teeth, many mammalian fossils, such as deers (Cervus sp.), macaques (Macaca sp.), and porcupines (Hystrix sp.), were also uncovered from Longshia-dong Cave. These mammalian fossils were once considered the kills brought back by leopards. Nevertheless, a previous study pointed out that the deposit in Longshia-dong Cave is a result of multiple reworkings (Wang, 2015). The co-occurrence of the mammalian fossils in Longshia-dong Cave probably represents a fossil accumulation over thousands of years.

Moreover, leopards left various bite marks on prey’s bones (Shi & Wu, 2011; Binford, 2014), but Lin (2017) examined fossil bones from Longshia-dong Cave and failed to find any common bite marks. Most deer fossils from the cave are mandibles or limb bones, all of which are unfavorable to leopards (Li, 2007). Thus, we suggest that the accumulation of mammalian fossils in Longshia-dong Cave is very likely a result of multiple reworkings.

Figure 10 Discovery sites of Pleistocene leopard fossils in East Asia.

Discovery sites of Pleistocene leopard fossils in East Asia. Early Pleistocene 1, Zhen’an, Shaanxi Province, China (Li & Deng, 2003); 2, Bailong Cave, Yunxi, Hubei Province, China (Wu et al., 2009); 3, Longgupou, Wushan County, Chongqing, China (Jin et al., 2008); 4, Cili County, Hunan Province, China (Wang et al., 1982); 5, Hezhang County, Guizhou Province, China (Zhao et al., 2013); 6, Yuanmou, Yunnan Province, China (You & Qi, 1973); Middle Pleistocene 7, Chaoxian, Anhui Province, China (Xu et al., 1984); 8, Yanjinggou (formerly Yenchingkou), Chongqing, China (Young, 1935); 9, Panlong Cave, Yunfu, Guangdong Province, China (Wang et al., 1990); Late Pleistocene 10, Niuyan Cave ( = Bull Eye Cave), Mentougou, Beijing, China (Deng, Huang & Wang, 1999); 11, Locality 1, Zhoukoudien (formerly Choukoutien), Beijing, China (Pei, 1934); 12, Locality 13, Zhoukoudien, Beijing, China (formerly Choukoutien) (Teilhard de Chardin & Pei, 1941); 13, Anyang, Henan Province, China (Teilhard de Chardin & Young, 1936); 14, Gongwangling, Lantian County, Shaanxi Province, China (Hu & Qi, 1978); 15, Yang’er Cave, Huatan County, Hunan Province, China (Wu, Deng & Zheng, 2008); 16, Fuyan Cave, Daoxian, Hunan Province, China (Li et al., 2013); 17, Mawoukou Cave, Bijie, Guizhou Province, China (Zhao et al., 2016); 18, Heshang Cave, Fumin County, Yunnan Province, China (but possibly Middle Pleistocene; (Zhang et al., 1989)); 19, Liucheng Gigantopithecus Cave, Guangxi Province, China (Pei, 1987); 20, this study. 21, Lower Hamakita Formation, Honshu Island, Japan (Suzuki, 1966) 22, Chongphadae Cave, Hwangju County, Democratic People’s Republic of Korea (Choe et al., 2020).

Table 3 Comparisons of dental size measurements (mm) of Longshia-dong Cave fossil to Chinese leopard fossils (expressed by length × width).

	Longshia-dong Cave (Kenting, Taiwan)	Niuyan Cave (Beijing, China)	Zhoukoudian (Beijing, China)	Gongwangling (Shaanxi, China)	recent	
		V11799	V11800	V11801	location1st	location 13th	V2980			
p3	12.5 × 6.39	13.4 × 6.5	–	–	16.3 × 9	16 × 7.5	14.4 × 8	14.8 × 7.9	12 × 6.3	
p4	17.49 × 8.51	18.2 × 9.7	18.5 × 9.5	–	23.2 × 12	21 × 10	21 × 11.2	22.3 × 11	18 × 9.4	
m1	16.77 × 7.95	19.3 × 9.2	19.8 × 8.5	19.5 × 8.5	24 × 12.2	21 × 10	22.2 × 12	22.7 × 12.3	18.3 × 8.6	

A comparison of the fossil teeth from Longshia-dong Cave to Chinese leopard fossils

Leopards are widely distributed throughout Asian and African continents, but only a few of them are currently present on islands such as Java and Sri-Lanka (Pocock, 1932). No written literature in Taiwan has reported the presence of leopards; however, this study reported the first leopard fossils from Longshia-dong Cave and thus suggested the presence of leopards in the Late Pleistocene of Taiwan.

Many previous studies suggested a continental origin of Taiwanese mammalian fossils based on various lines of evidence (Otsuka & Lin, 1984; Lai, 1989; Qi, Ho & Chang, 1999; Chen, 2000a; Fooden & Wu, 2001). The Late Pleistocene of China has produced numerous leopard fossils from various sites, including Beijing Mentougou Niuyan Cave (Deng, Huang & Wang, 1999), the first and thirteenth locations of Beijing Zhoukoudian Site (Pei, 1934), Shaanxi Lantian Gongwangling Site (Hu & Qi, 1978), Anyang Yinxu Site (Teilhard de Chardin & Young, 1936), and Guangxi Liucheng Cave (Pei, 1987) (Fig. 10). Among these reports, Deng, Huang & Wang (1999) claimed that the two lower first molars of the three fossil leopard teeth uncovered from Mentougou Niuyan Cave are the smallest (19.3 × 9.2 mm and 19.5 × 8.5 mm) in comparison with the other Chinese fossil leopard teeth. The size of the fossil teeth in our study (16.77 × 7.95 mm; Table 3), however, are much smaller than the teeth reported in Deng, Huang & Wang (1999). The smaller dental size can be explained by three hypotheses: (1) ontogenetic variation, (2) individual size variability, and (3) insular dwarfism. The first hypothesis is here precluded as both of our specimen and the Mentougou Niuyan Cave specimen are permanent teeth. The second hypothesis is currently a working hypothesis in the presence of only one fossil specimen. Despite the uncertainty arisen from the poor sample size (N=1), the third hypothesis is an interesting explanation. Meiri, Dayan & Simberloff (2005) indicated that the m1 size in carnivores randomly varies in different habitats, albeit not regularly and predictably with either area or isolation. They concluded three selective forces, including resource limitation, predation, and interspecific competition. Our study area, Kenting National Park, has produced less various fossils than all aforementioned Chinese sites, suggesting a habitat with fewer resources for carnivores. Besides, the leopard described in this study is the only carnivores found in Kenting National Park to date, potentially indicating lower predation pressure by other carnivores and the absence of interspecific competition. While the absence of interspecific competition would have resulted in gigantism, both of the fewer resources and lower predation pressure possibly contributed to the smaller size of Taiwanese leopards than Chinese leopards. Yet, more specimens are required for further studies and will help illuminate the smaller dental size of the Longshia-dong Cave specimen

Conclusions

Leopards are a group of carnivores widely distributed throughout Asian and African continents, yet they are no longer found in eastern Asia due to civilization. Fossils thus represent the only clue indicating prehistoric leopards. While their teeth are significantly smaller than the other leopard teeth compared and thus are easily identified, poorly preserved fossils hinder further investigations. This study, based on an integration of morphological and geometric morphometric analyses, reveals the assignment of the fossil teeth excavated from Longshia-dong Cave to Panthera pardus, which is currently absent on Taiwan Island and suggests the presence of leopards in the Late Pleistocene of Taiwan. Such a record of prehistoric leopards in Taiwan thus adds up to the carnivore biodiversity of Taiwan. However, whether the Panthera individual is aboriginal or migrated is still uncertain. Our study also found that the teeth from Longshia-dong Cave belonged to a smaller individual in comparison with the fossil record from China indicates the smaller size of Taiwanese leopard fossil teeth than Chinese ones. Such a smaller dental size was possibly a result of individual size variability or insular dwarfism. To conclude, the discovery of the leopard fossil in Taiwan opens shed some light on the origin of the Kenting Fauna. Nevertheless, the insular dwarfism of, and the migration history of, the prehistoric leopards in Taiwan are still speculations and thus require more specimens and studies.

Supplemental Information

Supplemental Information 1 Measurements of teeth in all specimen in mm

Click here for additional data file.

Supplemental Information 2 Quantities of the measurements

Click here for additional data file.

Supplemental Information 3 Factor loadings for selected ratios of p3, p4, and m1

Click here for additional data file.

The authors would like to thank Y-J Chen of NMNS, S-W Chang of the Endemic Species Research Institute, and T-C Chan of Taipei Zoo for their assistance in accessing the specimens. We are also grateful to P-J Chiang for his helpful comments and J Chen for her help on illustrations. Three anonymous reviewers and Martin Sabol have helped improve this study with their constrictive comments. Appreciation also goes to all members of the Taiwan-Japan Joint Excursion in 2014. We also thank C-Y Lin, K-C Wang and T-Y Hsiao for their assistance in the excursion. We also thank J-S Chi for proofreading.

Additional Information and Declarations

Competing Interests

Author Contributions

Data Availability

The authors declare there are no competing interests.

Tzu-Chin Chi conceived and designed the experiments, analyzed the data, prepared figures and/or tables, authored or reviewed drafts of the paper, and approved the final draft.

Yi Gan conceived and designed the experiments, performed the experiments, analyzed the data, prepared figures and/or tables, and approved the final draft.

Tzu-Ruei Yang analyzed the data, authored or reviewed drafts of the paper, and approved the final draft.

Chun-Hsiang Chang conceived and designed the experiments, authored or reviewed drafts of the paper, and approved the final draft.

The following information was supplied regarding data availability:

Raw measurements are available in the Supplemental Files.

Specimens are housed at the National Museum of Natural Science (NMNS) in Taichung, Taiwan. All the studied specimens (F056584, F056585, and F056586) have been accessioned into NMNS under the accession numbers provided.

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

Zhang et al. (1989) Zhang Y Long Y Ji H Ding S 1989 The Cenozoic deposits of the Yunnan region Professional Papers on Stratigraphy and Paleontology 7 1 21 (in Chinese)
Zhao et al. (2016) Zhao L Zhang L Du B Nian X Zheng Y Zhang Z Wang C Wang X Cai H 2016 New discovery of human fossils and associated mammal fauna from Mawokou Cave in Bijie, Guizhou Province of southern China Acta Anthropologica Sinica 35 24 35 (in Chinese with English abstract)
Zhao et al. (2013) Zhao L Zhang L Xu C Wang X Cai H 2013 New discovered Early Pleistocene mammal fossils and bone artifact in Hezhang of Guizhou Plateau Acta Anthropologica Sinica 32 477 484 (in Chinese with English abstract)