# Peer review of "First report of leopard fossils from a limestone cave in Kenting area, southern Taiwan"

_PeerJ, doi:10.7717/peerj.12020_

## Round 0.1 · original submission · Major Revisions

Thank you very much for submitted very interesting research. However, concluded from four reviewers, your manuscript needs major revisions.

For myself, I have an important point that how you sure that your sample (fossil) is leopard (Panthera pardus). Did you check the species by DNA sequencing? As we know that many samples in many museums were wrong species identification.

Reviewer 1 ·

Basic reporting

1) Language: the paper must undergo professional editing for English.
2) Literature cited: the manuscript requires many more references, especially in the methodological section, the taphonomic perspective in the discussion, and biogeography.
3) Field background: a plan and a section of the cave is required.
4) Why has the sediment not been dated, using 14C (for the overlying deposits), U-Th, or OSL?
5) Data summary and presentation is often faulty, see comments on pdf

Experimental design

The research is meaningful, as it potentially describes a new top predator taxon in a major location in Asia where it was not known before.
The investigation is not rigorous: I'm not sure that the GMM analysis (or the metric one) provide unequivocal taxonomic evidence that the teeth have belonged to a leopard. Consider using LDA or at least anova to separate groups. Also, zooMS and aDNA are basic in identifying new taxa today, especially when the morphological analysis is not equivocal.

The digitization process of the GMM is not clear: add a figure.

The pervalence of morphological criteria in each species is not clear.

Validity of the findings

Statistics are far from being state-of-the-art. The authors would benefit from a more thorough treatment of both their metric and GMM data.

Annotated reviews are not available for download in order to protect the identity of reviewers who chose to remain anonymous.

Reviewer 2 ·

Basic reporting

The present manuscript tried to identify the species of the fossil felid teeth from Kenting area. The morphological features were well described and combined tests including linear measurements and geometric morphometric methods were well done. I believe this manuscript is worth publishing after minor revision for several points.

Experimental design

There may be other ways that provide more robust evidence (such using discriminant analysis or comparison of Procrustes distances etc.), but the study design is enough to the conclusion of the present manuscript.

Validity of the findings

The authors used several different methods to distinguish the taxon of the fossils. Accordingly, I consider the result is valid enough.

Additional comments

Major points:
The term “geomorphometry” possesses several different meanings. I suggest using “geometric morphometrics” which is the special term indicating landmark-based morphometrics.

When comparing extant and fossil specimens (or bite marks in fossil bones), it might be cautious that clouded leopard teeth (especially canine) have been evolving at a very fast rate (Harano and Kutsukake, 2018). However, I feel a discussion about it is not essential.

Minor points:

Line 50: I am afraid that “Ai Kawamura” is a mistake of “Yoshinari Kawamura”. Ai did not belong to Aichi University of Education (She belonged at Osaka City University during her PhD course and moved to Toyama University). Otherwise, affiliation and prefix may be changed.

Line 93: I think this sentence needs a citation.

Line 135: “TPS software” should be clearly explained as TPS Dig etc.

Line 150-151: This explanation seems unclear. It might be better to explain it clearly.

Reviewer 3 ·

Basic reporting

-

Experimental design

-

Validity of the findings

-

Additional comments

The statistical analysis of the study is appropriate however some points need to be clarified and clearly described.
1. The authors indicated that the all PCA in this study were performed using R (line 149-150) thus it is not necessary to mention about R in line 157, 163, 165 with similar sentences.

2. A brief of the advantage of using two rounds of PCA should be described in term of statistical concepts as it will help some audiences who are not familiar with such technique to understand.

3. Line 181. Do not need to address Fig. 6 as this figure is the result.

4. Line 230. It is not necessary to address “see Material and methods”. This term should be removed.

5. The author showed Fig 9 but no description for the construction of bivariate plots and also the regression analysis. These analyses should be described in MM. Results from analyses should be included in the results sections.

·

Basic reporting

The submitted manuscript yields very important data on the occurrence of leopards in Taiwan during the Late Pleistocene, probably during the Last Glacial Period, when Taiwan was connected with the continental Asian part. Although it is so far an isolated fossil record of juvenile individual, it is a clear evidence for migration routes of these felis cats and their biogeographical distribution in the past. From this point of view, the presented data are very fundamental for our knowledge of the Quaternary megafauna.
Since I am not native speaker, I am not able to consider the level of used professional English, but I think it is clear, unambiguous, and technically correct, even though there are a few imperfections (I think) in some places (see attachment).
The used literature references are relevant and appropriately reffered in the vast majority of cases. I only found cited works of Pocock (1917) and Oken (1816) in the "systematic part" but not listed in the "References" section, whereas other two works (Fischer de Waldheim, 1817 and Linnaeus, 1758) are reffered there... (see attachment).
The article structure, figures, and tables are professional and relevant, meeting the standard for this type of publication. Raw data are shared and their analyses support the obtained results. Based on these, two hypotheses are propounded. For comments to their relevance, see attachment.

Experimental design

The manuscript presents an original primary research. The main research questions are well defined, with the relevant introduction to the problem of study. The investigation meets all scientific and ethical standards. The analythic methods described are appropriate, producing relevant proof data and results that are also useful for another investigators.

Validity of the findings

All obtained data and results of all used statistic analyses support the attribution of found felid teeth to Panthera pardus. Based on that, I do not understand why authors in some place abandon this correct assignation and write "we are only able to assign them to the subfamily of Pantherinae", when the determination of fossil record under study is unambiguously supported by metrics, morphology, and statistics.
The conclusions, in my opinion, need major revision. The hypothesis (1) could be focused rather on the individual variability than on the ontogenetic one and the hypothesis (2) is rather speculation, since uncovered teeth are only so far isolated fossil record, too small for the thinking about a dwarfism, especially when Taiwan was probably a part of continental Asia minimally during the Last Glacial ... In spite of that, I think the article should be accepted.

Additional comments

Although isolated, but very important fossil record of leopard in Pleistocene of Taiwan is supported by well realized morphometric and statistic analyses. However, hypothetical impact of this find should be re-interpreted. All my comments and remarks are highlighted directly in the attached text of manuscript.

---

## Round 0.2 · Minor Revisions

One review had some comments for improving your manuscript. I will accept after your revision on this round.

Reviewer 2 ·

Basic reporting

I feel all concerns I had pointed out were corrected. I have no further comments.

Experimental design

The design is enough to the present study.

Validity of the findings

I feel the result is valid enough.

Additional comments

I have no additional comments.

Reviewer 3 ·

Basic reporting

no comment

Experimental design

no comment

Validity of the findings

no comment

·

Basic reporting

no comment

Experimental design

no comment

Validity of the findings

no comments

Additional comments

I found only a few failures in this re-worked manuscript, all highlighted in the reviewed file.

---

## Round 0.3 · accepted · Accept

Congratulation on your publication.